# Mother and newborn skin to skin contact in East Africa: Prevalence and predictors (2019–2023)

**Alemneh Tadesse Kassie** [1]*, Tadesse Tarik Tamir[2], Astewil Moges Bazezew[3], Agnche Gebremichael[4], Daniel Asefa Gonete[5], Alebachew Ferede Zegeye[6]

1 Department of Clinical Midwifery, School of Midwifery, College of Medicine and Health Sciences, University of Gondar, Gondar, Ethiopia, 2 Department of Pediatrics and Child Health Nursing, School of Nursing, College of Medicine and Health Sciences, University of Gondar, Gondar, Ethiopia, 3 Department of Surgical Nursing, School of Nursing, College of Medicine and Health Sciences, University of Gondar, Gondar, Ethiopia, 4 Department of General Midwifery, School of Midwifery, University of Gondar College of Medicine and Health Science, Gondar, Ethiopia, 5 Department of public health Pawi Health Science College, Benishangu Gumuz Region, Northwest, Ethiopia, 6 Department of Medical Nursing, School of Nursing, College of Medicine and Health Sciences, University of Gondar, Gondar, Ethiopia

* alexotadesse.79@gmail.com

## Abstract

Skin-to-skin contact (SSC) between newborns and mothers is crucial for reducing hypothermia complications and enhancing mother-baby bonding. This study aimed to examine the prevalence rates and risk factors for SSC in five East African countries. Data from the most recent Demographic and Health Surveys covering five East African countries from 2019 to 2023 were utilized for secondary data analysis. A total of 37,140 newborns were included in the weighted sample sizes. The factors contributing to SSC prevalence were assessed using a multilevel mixed-effects logistic regression model, with significance declared at p-values <0.05. Adjusted odds ratios (AOR) and confidence intervals (CI) were used for interpretation. In East Africa, nearly half of newborns did not receive SSC. Individual-level factors associated with SSC included: immediate breastfeeding (IBF) (AOR = 2.24, 95% CI: (1.86, 2.69)), primary education (AOR = 1.60, 95% CI: (1.24, 2.07)), more than four antenatal care visits (AOR = 2.15, 95% CI: (1.37, 3.36)), low birth weight (AOR = 1.35, 95% CI: (1.04, 1.76)), multiple births (AOR = 0.41, 95% CI: (0.24, 0.73)), and cesarean section deliveries (AOR = 0.26, 95% CI: (0.20, 0.34)). Community-level factors included rural residence (AOR = 0.63, 95% CI: (0.48, 0.82)), with significant country-level variations: Kenya (AOR = 2.45, 95% CI: (1.72, 3.49)), Mozambique (AOR = 1.21, 95% CI: (0.91, 1.62)), and Tanzania (AOR = 3.14, 95% CI: (2.29, 4.31)). The findings indicate a low prevalence of mother-newborn SSC in East African countries, with particularly high rates observed in Rwanda and lower rates in Madagascar. Policymakers and ministries of health should prioritize initiatives aimed at improving SSC, particularly for newborns and mothers with low educational levels, limited media exposure, and those living in rural areas, to enhance maternal and neonatal care during delivery.

**Data availability statement:** Third party data was obtained for this study from DHS Program. Data may be requested from DHS Program after creating an account and submitting a concept note. More access information can be found on the DHS Program website (https://dhsprogram.com/data/Access-Instructions.cfm). The authors confirm that interested researchers would be able to access these data in the same manner as the authors. The authors also confirm that they had no special access privileges that others would not have.

**Funding:** The author(s) received no specific funding for this work.

**Competing interests:** The authors declared that there is no competing interest.

## Introduction

Skin-to-skin contact (SSC) involves placing the naked baby on the mother's chest between her breasts, promoting bonding immediately after birth [1,2]. This practice has well-documented benefits, including earlier expulsion of the placenta, reduced bleeding, increased breastfeeding self-efficacy, and lower maternal stress [3]. The rise in oxytocin during the first hour after-birth is linked to mother-infant bonding [4]. skin-to-skin contact (SSC) should ideally start at birth and continue until the first breastfeeding, fostering neuro-behaviors that meet basic biological needs and potentially programming future physiology and behavior during this sensitive period [5].

The first hour after birth, known as the sensitive period, is the optimal time for mothers to initiate breastfeeding. During this time, even brief maternal-infant separation can hinder the neonate's ability to start breastfeeding. However, the effects of skin-to-skin contact (SSC) on both the mother and infant beyond the newborn period, as well as the evolving mother-infant relationship, remain less researched [6]. The WHO recommends that skin-to-skin contact should begin the first hour after birth; ideally, it starts immediately after delivery and can continue for up to 23 hours [7].

Evidence indicates that approximately two-thirds of neonatal deaths occur within the first day, with over three-quarters happening in the first week of life. Every year, 2.5 million neonates die, mostly in low- and middle-income countries [8]. The African region has the highest neonatal mortality rate at 28.0 per 1,000 live births, followed by the Eastern Mediterranean (26.6) and South-East Asia (24.3) [9]. In 2019, Sub-Saharan Africa accounted for 42% of global neonatal deaths, with a mortality rate of 27 deaths per 1,000 live births [10]. Neonatal mortality rates in East Africa at 33.6 deaths per 1,000 live births, addressing the factors contributing to these high rates is crucial for enhancing maternal and neonatal care in this area [9]. Notably, it is the only Sustainable Development Goal (SDG) region that has not seen a decline in neonatal deaths since 1990 [11,12].

Neonatal mortality in East Africa remains a significant public health challenge, driven by factors such as inadequate healthcare infrastructure, poverty, infections, and limited access to quality maternal and neonatal care [13]. One effective intervention to reduce neonatal mortality is SSC, or kangaroo care [14]. This practice offers benefits like temperature regulation, improved breastfeeding initiation, enhanced bonding, and increased survival rates. In East Africa, neonatal mortality is significantly associated with home births, lack of maternal education, and decisions about contraceptive practices made solely by husbands, even when controlling for rural residency [9]. Breastfeeding within the first hour after birth is vital for newborn survival, providing essential nutrients, immunological protection, and fostering emotional bonding between mother and child. SSC plays a significant role in facilitating early breastfeeding by stimulating the newborn's rooting and sucking reflexes [15,16]. However, many regions in East Africa face challenges that delay or prevent this critical practice.

Newborns are particularly vulnerable to temperature fluctuations, which can lead to hypothermia a leading cause of neonatal death, especially in low-resource settings. SSC is crucial for regulating a newborn's body temperature [17], reducing the risk of

cold stress and hypothermia. Unfortunately, the low utilization of SSC in many areas leaves newborns at risk, particularly in rural regions where incubators and other thermal support are not readily available [18]. SSC supports the newborn's immune system, without adequate SSC, the risk of infections increases, particularly in situations where hygiene practices are poor or breastfeeding is delayed [18,19]. Early bonding is vital not only for physical health but also for the emotional and psychological well-being of both mother and newborn.

Limited awareness and education about the importance of SSC contribute to its low prevalence. Healthcare providers may not prioritize this practice or may fail to communicate its benefits effectively to new mothers. Research indicates that kangaroo mother care, which incorporates SSC, can reduce neonatal mortality by up to 40%, particularly in low- and middle-income countries [20]. Enhancing SSC practices is crucial for reducing neonatal mortality in low- and middle-income countries, particularly in this region. Effective health interventions can significantly improve newborn health outcomes, making it vital to understand and address the barriers to SSC in order to promote better maternal and infant health practices. This research aims to assess the prevalence of SSC among newborns in East Africa and identify associated factors.

## Methods

### Study design, study area, and period

A population-based cross-sectional study was conducted using recent data from the Demographic and Health Surveys (DHS) across five East African countries, collected between 2019 and 2023. This study employed multilevel mixed-effects analysis to assess various health and health-related indicators. The DHS is a community-based survey conducted every five years, providing updated insights into health metrics and trends within the region.

### Data source, study population and sampling technique

The latest Demographic Health Survey (DHS) datasets from East African countries, covering the years 2019 to 2023, were used for this secondary data analysis, focusing on five countries: Kenya, Madagascar, Mozambique, Rwanda, and Tanzania. The DHS datasets are known for their strong validity, achieved through standardized data collection methods and comprehensive questionnaires designed to capture relevant health indicators. This standardization facilitates accurate comparisons across countries and populations. By utilizing these validated datasets, the study effectively evaluates the prevalence and contributing factors of skin-to-skin contact (SSC) among newborns.

The reliability of the DHS data is enhanced by a stratified two-stage cluster sampling design, which reduces selection bias and ensures that the samples are representative of the populations studied. Additionally, rigorous quality control measures are implemented during data collection and processing, improving the consistency and dependability of the data. This reliability is essential for drawing meaningful conclusions in the analysis of factors influencing SSC.

For this analysis, the variable "was the child placed on the mother's chest and bare skin after birth" (m77) was recoded to establish the outcome variable for SSC. A binary logistic regression model was employed to identify factors associated with SSC, with adjusted odds ratios (AOR) reported at a significance level of 95%. Variables with a p-value of <0.25 were considered for multivariable analysis, while those with p-values <0.05 were regarded as statistically significant. The total weighted sample of 37,140 children provides a robust foundation for analyzing factors influencing SSC among newborns in the selected East African countries "Table 1."

### Study variables

**Dependent variables.** In this study, the dependent variable was skin-to-skin contact (SSC). It was categorized as "Yes = 1" if SSC occurred on the mother's abdomen or between her breasts. Conversely, it was recorded as "No = 0" if

**Table 1. Sample size for Individual and community-level determinants of skin to skin contact among new-borns in the East African countries, DHS 2019-2023.**

| Country | Year of survey | Weighted sample (n) | Weighted sample (%) |
|---|---|---|---|
| Kenya | 2022 | 5,585 | 15.04% |
| Madagascar | 2021 | 12,499 | 33.65% |
| Mozambique | 2022/23 | 5,139 | 13.84% |
| Rwanda | 2019/20 | 8,092 | 21.79% |
| Tanzania | 2022 | 5,825 | 15.68% |
| Weighted sample size | | 37,140 | 100% |

**DHS:** Demographic Health Survey.

skin-to-skin contact was not performed. This binary classification allowed for a clear assessment of the prevalence and factors associated with SSC among newborns in the selected East African countries.

## Independent variables

In this analysis, independent variables were sourced from two levels—individual and community—reflecting the hierarchical nature of the DHS data.

**Individual-level variables.** The individual-level independent variables included maternal age, categorized as 15–19, 20–24, 25–29, 30–34 and 35–49 years. Initiation of first Breast feeding within one hour after birth yes or no, Maternal education was assessed at various levels, including no formal education, primary, secondary and higher education. Marital status was defined as either "in union" or "currently non-union." The sex of the child was noted as male or female, while birth weight was classified as low or normal. Birth order either first or second and above. The Pregnancy duration preterm or term. Antenatal checkups were categorized into no visits, fewer than four visits, or four or more visits. Actual number of baby delivered Single or multiple. Household media exposure yes, or no. Finally, the household wealth index was classified as poor, middle, or rich.

**Community-level variables.** The community-level independent variables included the place of residence, categorized as urban or rural. Community-level poverty was also categorized as low or high level poverty. Additionally, community media exposure was classified as low or high level of media exposure. The country category differentiated respondents based on their specific East African country names. Data leveling of both the dependent and independent variables was conducted using conceptual stratification, informed by existing literature and known facts relevant to the study's objectives. This approach ensured that the analysis captured a comprehensive range of factors influencing SSC contact among newborns.

## Data processing and statistical analysis

The data obtained from recent Demographic Health Survey (DHS) datasets were cleaned, recorded, and analyzed using STATA version 17 statistical software. The DHS data variables are organized into clusters, where observations within a cluster are more similar to each other than to those in other clusters. This clustering can violate the assumptions of independent observations and equal variance across clusters, highlighting the need for a more sophisticated analytical approach to account for between-cluster factors.

To address this issue, multilevel mixed-effects logistic regression was employed to determine the factors associated with SSC. This method follows a structured model approach consisting of four models. The Null Model includes only the outcome variable, allowing for the assessment of the variability of SSC across the clusters. Model I incorporates only individual-level variables, facilitating the evaluation of their association with the outcome. Model II includes only community-level variables, examining their relationship with SSC. Finally, Model III integrates both individual and

community-level variables simultaneously with the outcome variable (skin-to-skin contact). By analyzing these models, the study effectively assessed the associations of both individual and community characteristics with SSC, providing a comprehensive understanding of the factors influencing this important practice among newborns.

### Ethical approval and consent to participate

After the consent manuscript was submitted to the DHS Program for downloading the dataset for this investigation, the International Review Board of the Demographic and Health Surveys (DHS) program waived the requirement for informed consent. Since the study utilized data from a publicly accessible source, it was not classified as an experimental study. All methods were conducted in accordance with the principles outlined in the Helsinki Declaration. Further details regarding DHS data and ethical standards are available online at the DHS website (http://www.dhsprogram.com).

### Random effects

Random effects, or measures of variation for the outcome variables, were estimated using the median odds ratio (MOR), intra-class correlation coefficient, and proportional change in variance (PCV). The ICC and PCV were computed to assess the variation between clusters. By treating clusters as random variables, the ICC quantifies the variation in SSC rates across clusters, calculated as follows; $\text{ICC} = \dfrac{VC}{VC + 3.29} \times 100\%$.

The Median Odds Ratio (MOR) is the median value of the odds ratio which quantifies the variation or heterogeneity in SSC prevalence rates between clusters in terms of odds ratio scale and is defined as the median value of the odds ratio between the cluster at high likelihood of SSC at birth prevalence rates and cluster at lower risk when randomly picking out individuals from two clusters; $\text{MOR} = e^{0.95\sqrt{VC}}$ Moreover, the PCV demonstrates the variation in the SSC prevalence rates explained by determinants and computed as; $\text{PCV} = \dfrac{Vnull - Vc}{Vnull} \times 100\%$; where V-null is the variance of the null model and VC is the cluster-level variance. Fixed effects were utilized to estimate the association between the likelihood of SSC and both individual and community-level independent variables.

The strength of these associations was presented using adjusted odds ratios (AOR) and 95% confidence intervals, with a significance threshold set at $p < 0.05$. Due to the nested nature of the model, the deviance statistic ($-2$ log likelihood ratio) was employed to compare models, selecting the one with the lowest deviance as the best fit. Additionally, the variables included in the models were checked for multi-collinearity by measuring the variance inflation factors (VIF), with results falling within acceptable limits of 1–10, ensuring the robustness of the analysis [21].

## Results

### Socio demographic and economic characteristics of study participants

A total of 37,140 newborns were included in the analysis of this study. Nearly one-fourth of participant's mother were between the ages of 20 and 24, one third newborns were not breast feed within one hour after delivery. Regarding the educational status of the participant's mother nearly half of them gate primary school, and more than eighty percent of mother's marital status live in-union. Nearly two third of families have media exposure, around two third of mothers had > 4 times of ANC visit, nearly 95% of babies are term infants on their gestational age, participants' wealth index indicates that more than 45% of are poor. More than 85% of participants had the first birth. and about three-fourth (74.87%) were residents of rural areas of East Africa "Table 2."

### Prevalence of mother and newborn skin to skin contact in East Africa

The overall prevalence of mother and newborn skin to skin contact in East Africa was 51.70% at a CI of 95% (51.19,52.21). In rural areas, 48% (13,350) of newborns had SSC, compared to 62.7% (5,853) in urban areas (Fig 1). The

**Table 2. Socio-demographic and economic characteristics of study participants.**

| Individual level variables | Category | Frequency (n) | Percent (%) |
|---|---|---|---|
| Maternal Age In years | 15-19 | 3,104 | 8.36% |
| | 20-24 | 9,112 | 24.53% |
| | 25-29 | 8,881 | 23.91% |
| | 30-34 | 7,284 | 19.61% |
| | 35-49 | 8,759 | 23.58% |
| Initiation of first Breast feeding within one hour after birth | No | 12,283 | 33.07% |
| | yes | 24,857 | 66.93% |
| Maternal education | No formal education | 7,767 | 20.91% |
| | Primary | 17,982 | 48.42% |
| | Secondary and above | 11,391 | 30.67% |
| Maternal marital status | Non union | 6,744 | 18.16% |
| | In union | 30,396 | 81.84% |
| Mode of delivery | Abdominal | 33,676 | 90.67% |
| | Vaginal | 3,464 | 9.33% |
| Birth weight | Low | 3,581 | 9.64% |
| | Normal | 33,559 | 90.36% |
| Baby sex | Male | 18,878 | 50.83% |
| | Female | 18,262 | 49,17% |
| Actual number of baby delivered | Single | 36,374 | 97.94% |
| | Multiple | 766 | 2.06% |
| Household media exposure | No | 13,735 | 36.98% |
| | Yes | 23,405 | 63.02% |
| ANC visit | No visit | 2,471 | 6.65% |
| | < 4 times | 10,644 | 28.66% |
| | >/=4 times | 24,025 | 64.69% |
| Pregnancy duration (gestational age) | preterm | 1,417 | 3.82% |
| | Term | 35,152 | 94.65% |
| | Post term | 571 | 1.54% |
| Household Wealth index | Poor | 16,846 | 45.36% |
| | Middle | 7,123 | 19.18% |
| | Rich | 13,171 | 35.46% |
| Birth order | First | 32,031 | 86.24% |
| | Second and above | 5,109 | 13.76% |
| **Community level variables** | | | |
| Household Residence | Urban | 9,335 | 25.13% |
| | Rural | 27,805 | 74.87% |
| Community level of media exposure | Low media exposure | 10,659 | 28.70% |
| | High media exposure | 26,481 | 71.30% |
| Community level of poverty | Low poverty | 13,011 | 35.03% |
| | High poverty | 24,129 | 64.97% |

(ANC: Antenatal Care).

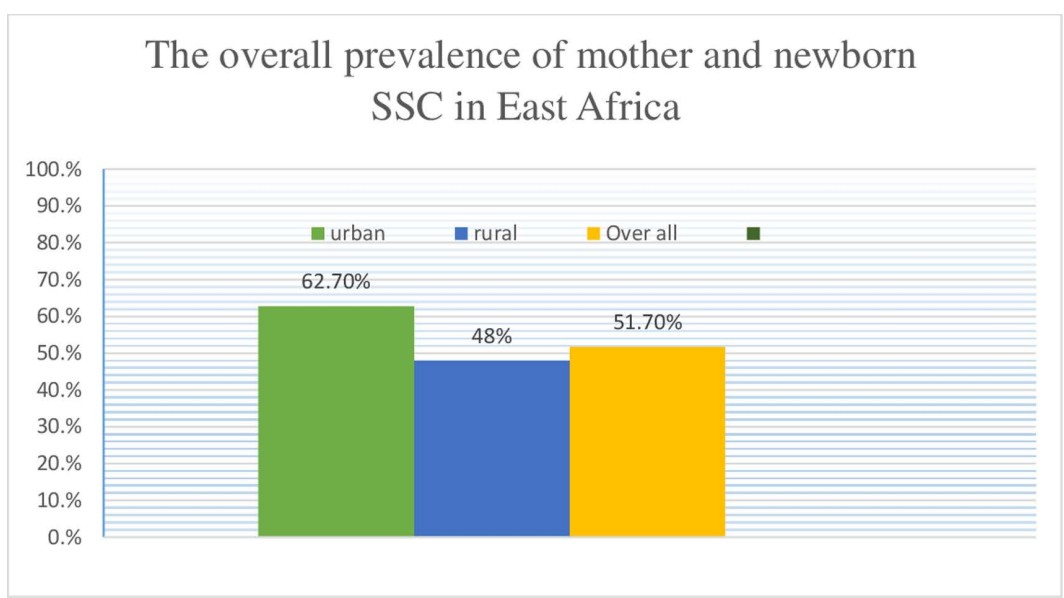

**Fig 1. The overall prevalence of mother and newborn SSC in East Africa, urban and rural areas.**

prevalence of SSC among newborns significantly varied across East African countries. Subsequently, the lowest prevalence of SSC was observed in Madagascar (23.13%), while the highest was in Rwanda (76.13%) (Fig 2).

### Random effect (Measures of variation) and model fitness

Findings from the null model revealed significant differences in the prevalence of skin-to-skin contact (SSC) between communities, with a variance of 0.5403876 and a p-value of 0.000. The variance within clusters accounted for 85.89% of the

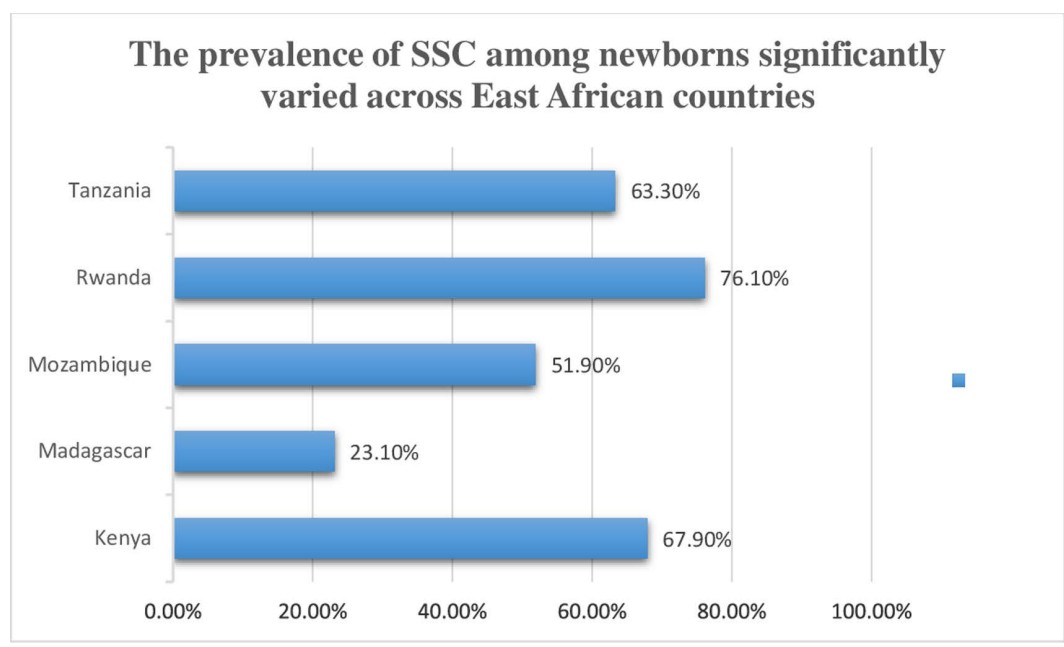

**Fig 2. The prevalence of SSC among newborns significantly varied across East African countries.**

total variation in SSC among newborns, while variance across clusters contributed 14.11%. The null model indicated that the odds of SSC in newborns varied between higher and lower risk clusters by a factor of 2.01 times. The intra-class correlation value for Model I indicated that 8.97% of the variation in skin-to-skin contact (SSC) prevalence could be attributed to disparities between communities. In Model II, which included community-level variables, cluster variations accounted for 5.47% of the differences in SSC prevalence rates, as reflected in the ICC value. In the final model (Model III), approximately 80.96% of the variation in the likelihood of SSC prevalence was attributed to both individual and community-level factors. The likelihood of SSC among newborns varied by a factor of 1.36 across low and high SSC newborn clusters. Model fitness was evaluated through log-likelihood ratios and deviance, with Model III emerging as the best-fitted model due to its lowest deviance (3,508.581) and highest log-likelihood ratio (-1,754.2905) "Table 3."

## Association of individual and community-level determinants and SSC prevalence among newborns in East African countries

In a multivariable multilevel mixed-effect logistic regression analysis that simultaneously examined both individual and community-level factors, several significant associations with skin-to-skin contact (SSC) were identified. Key individual-level variables included the initiation of breastfeeding within one hour after birth, maternal education, birth weight, number of babies delivered, household media exposure, and antenatal care visits. The analysis revealed that the odds of SSC were 2.24 times higher among newborns who did not receive breastfeeding within the first hour compared to those who did (AOR = 2.24, 95% CI: 1.86, 2.69). Additionally, newborns of mothers with only primary education had 1.60 times higher odds of SSC compared to those whose mothers had no formal education or secondary education and above (AOR = 1.60, 95% CI: 1.24, 2.07). Moreover, the odds of skin-to-skin contact (SSC) were 2.15 times higher for newborns whose mothers had four antenatal care visits compared to those with fewer or no visits (AOR = 2.15, 95% CI: 1.37, 3.36). Newborns with low birth weight had a 1.35 times higher likelihood of SSC compared to those with normal weight (AOR = 1.35, 95% CI: 1.04, 1.76). Conversely, the odds of SSC decreased by 59% for multiple births compared to single births (AOR = 0.41, 95% CI: 0.24, 0.73), while cesarean section deliveries were associated with a 74% lower likelihood of SSC compared to vaginal births (AOR = 0.26, 95% CI: 0.20, 0.34). Additionally, the odds of SSC were 1.62 times higher for newborns whose mothers had media exposure compared to those without (AOR = 1.62, 95% CI: 1.31, 2.00).

At the community level, living in a rural area was associated with a 37% lower likelihood of skin-to-skin contact (SSC) compared to urban residence (AOR = 0.63, 95% CI: 0.48, 0.82). There were also significant country-level variations in SSC prevalence, with Kenya exhibiting 2.45 times higher odds of SSC compared to Madagascar (AOR = 2.45, 95% CI: 1.72, 3.49). Furthermore, newborns in Rwanda and Tanzania had notably higher odds of SSC, with AORs of 7.23 (95% CI: 5.41, 9.67) and 3.14 (95% CI: 2.29, 4.31), respectively, when compared to Madagascar "Table 4."

**Table 3. Model comparison and random effect analysis for skin to skin contact prevalence rates among new-born babies in the East Africa countries.**

| Parameter | Null model | Model I | Model II | Model III |
|---|---|---|---|---|
| variance | 0.5403876 | 0.3242507 | 0.1904324 | 0.1028669 |
| ICC | 14.11% | 8.97% | 5.47% | 3.03% |
| MOR | 2.01 | 1.72 | 1.51 | 1.36 |
| PCV | Reference | 39.99% | 64.76% | 80.96% |
| **Model fitness** | | | | |
| LLR | -24,960.903 | -22,869.234 | -1,925.4382 | -1,754.2905 |
| Deviance | 49,921.806 | 45,738.468 | 3,850.8764 | 3,508.581 |

(**ICC**: Intra-cluster Correlation, **LLR**: Log-likelihood Ratio, **MOR:** Median Odds Ratio, **PCV**: Proportional Change in Variance).

**Table 4. Multivariable multilevel logistic regression analysis of individual-level and community level factors associated skin to skin contact prevalence rates among new-borns East Africa, DHS (Demographic Health Survey) 2019-2023.**

| Individual-level variables | | Model I (AOR=95%) | Model II (AOR=95%) | Model III (AOR=95%) |
|---|---|---|---|---|
| Maternal Age In years | 15-19 | 0.63(0.57,0.69) | | 0.96(0.66,1.38) |
| | 20-24 | 0.76(0.71,0.81) | | 0.95(0.66,1.36) |
| | 25-29 | 0.93(0.87,0.99) | | 1.33(0.92,1.92) |
| | 30-34 | 0.97(0.90,1.04) | | 1.19(0.82,1.37) |
| | 35-49 | 1 | | 1 |
| Initiation of first Breast feeding within one hour after birth | No | 1 | | 1 |
| | yes | 2.53(2.40,2.66) | | **2.24(1.86,2.69)*** |
| Maternal education | No formal education | 1 | | 1 |
| | Primary | 1.45(1.36,1.55) | | **1.60(1.24,2.07)*** |
| | Secondary and above | 1.55(1.43,1.67) | | 1.24(0.90,1.69) |
| Maternal marital status | Non-union | 0.96(0.90,1.01) | | 1.19(0.95,1.50) |
| | In-union | | | 1 |
| Mode of delivery | Caesarian | 0.52(0.48,0.56) | | **0.26(0.20,0.34)*** |
| | Vaginal | 1 | | |
| Birth weight | Low | 1.68(1.55,1.82) | | **1.35(1.04,1.76)*** |
| | Normal | 1 | | 1 |
| Baby sex | Male | 1 | | 1 |
| | Female | 1.01(0.96,1.05) | | 0.93(0.78,1.09) |
| Actual number of baby delivered | Single | 1 | | 1 |
| | Multiple | 0.76(0.65,0.90) | | **0.41(0.24,0.73)*** |
| Household media exposure | No | 1 | | 1 |
| | Yes | 1.80(1.71,1.90) | | **1.62(1.31,2.00)*** |
| ANC visit | No visits | 0.47(0.43,0.53) | | 0.76(0.48,1.20) |
| | < 4 times | 1 | | 1 |
| | >/=4 times | 1.10(1.04,1.16) | | **1.62(1.33,1.98)*** |
| Pregnancy duration | preterm | 0.91(0.73,1.13) | | 0.96(049,1.88) |
| | Term | 1.38(1.15,1.67) | | 0.91(0.49,1.66) |
| | Post term | 1 | | 1 |
| Household Wealth index | Poor | 0.88(0.82,0.93) | | 0.98(0.75,1.28) |
| | Middle | 1 | | 1 |
| | Rich | 1.18(1.10,1.26) | | 1.10(0.83,1.47) |
| Birth order | First | 1 | | |
| | Second and above | 0.56(0.52,0.60) | | 0.75(0.57,1.00) |
| **Community level variables** | | | | |
| Household Residence | Urban | | 1 | 1 |
| | Rural | | 0.60(0.48,0.74) | **0.63(0.48,0.82)*** |
| Community level of media exposure | Low exposure | | 0.83(0.60,1.13) | 0.69(0.51,0.94) |
| | High exposure | | 1 | 1 |
| Community level of poverty | Low poverty | | 1 | 1 |
| | High poverty | | 0.85(0.62,1.15) | 0.82(0.63,1.01) |
| Country region | Kenya | | 1.49(1.08,2.05) | **2.45(1.72,3.49)*** |
| | Madagascar | | 1 | 1 |
| | Mozambique | | 0.99(0.78, 1.27) | 1.21(0.91,1.62) |
| | Rwanda | | 6.36(4.93,8.21) | **7.23(5.41,9.67)*** |
| | Tanzania | | 2.58(1.94,3.41) | **3.14(2.29,4.31)*** |

(ANC: Antenatal Care, AOR: Adjusted Odds Ratio).

## Discussion

Despite years of maternal and child health services, neonatal mortality remains a critical issue in low- and middle-income countries, particularly in East Africa. Many newborns still lack access to essential care, including skin-to-skin contact at birth. This study investigated the individual and community-level correlates of SSC among newborns in East Africa, revealing a prevalence of 51.7% (95% CI: 51.19, 52.21). This finding is concerning, as SSC is known to enhance neonatal health outcomes, including thermal regulation, bonding, and breastfeeding initiation. This finding is also lower than the previous studies conducted in Vietnam 88.7% and United States 74% [23,24]. Conversely the lower prevalence of SSC in this study compared to findings in Vietnam and the U.S. may be due to differences in socio-economic status, health infrastructure, and cultural practices. Health service coverage and maternal healthcare quality are generally more robust in those countries. Additionally, healthcare providers in East Africa may be inattentive to cost-free practices like SSC, which could contribute to its limited implementation. Addressing these disparities is essential for improving neonatal care and health outcomes for newborns in the region. Conversely, the prevalence of SSC in this study was lower than findings from Nigeria 13% [25], Ethiopia 48% [26], and Bangladesh 26% [27]. These discrepancies may stem from using secondary data from the Demographic Health Surveys (DHS) across five East African countries, while other studies focused on single-country data. Varied health infrastructure and maternal healthcare quality in East Africa influence SSC rates, making it essential to address these differences for improving neonatal care. In the multivariable multilevel mixed effect logistic regression analysis, significant factors associated with mother and newborn SSC included initiation of breastfeeding within one hour, maternal education, birth weight, number of babies delivered, mode of delivery, household media exposure, ANC visits, household residence, and country category (Kenya, Rwanda, and Tanzania).

In this study, the odds of initiating breastfeeding within the first hour were 2.24 times higher among newborns who experienced SSC compared to those who did not (AOR = 2.24, 95% CI: 1.86, 2.69). This finding aligns with previous research [25]. While the initiation of breastfeeding within one hour is around 67%, the lower prevalence of SSC indicates that mothers may lack knowledge about its benefits when combined with breastfeeding. This gap highlights the need for educational interventions to promote both practices for better neonatal health outcomes.

Women with primary education had 1.6 times higher odds of experiencing SSC compared to those with no formal education, indicating that maternal education plays a crucial role in neonatal care practices. Educated mothers are more likely to access and understand health-related information, engage with healthcare services, and follow recommended newborn care guidelines, leading to higher SSC rates.

The odds of practicing SSC were 1.35 times higher among mothers of low birth weight children. This finding mirrors results from related studies [11], suggesting a heightened awareness among these mothers regarding the importance of SSC for their vulnerable newborns. Mothers of low birth weight infants may be more inclined to engage in SSC as a means of promoting thermal regulation, bonding, and improved breastfeeding outcomes. This highlights the need for targeted support and education for mothers of low birth weight infants to encourage SSC practices and enhance neonatal care.

In this study, the odds of practicing SSC were 2.15 times higher among women who received optimal antenatal care during pregnancy compared to those who has < 4 ANC visits. This finding is consistent with previous research [28,29]. indicating that Optimal ANC likely provides women with essential information and support, fostering practices that promote early bonding and improved neonatal outcomes.

The odds of practicing SSC among women who had a cesarean delivery were 74% lower compared to those who had a vaginal delivery. This finding is consistent with similar studies [11]. Which suggest that the nature of delivery can significantly impact the initiation and practice of SSC. Challenges in recovering from cesarean sections may hinder immediate skin-to-skin contact, highlighting the need for strategies to promote SSC even after surgical delivery. Addressing these barriers is crucial for enhancing paternal-infant bonding and improving neonatal health outcomes.

The odds of practicing SSC were 59% lower among women who experienced multiple births compared to those with single births. This finding highlights the unique challenges that multiple birth mothers may face, such as increased physical demands and the need for additional support. The odds of practicing SSC were found to be 1.62 times higher among women who had media exposure compared to those who did not, as supported by study

[22,28]. This finding suggests that exposure to health-related media can significantly enhance awareness and knowledge about the benefits of SSC. Media campaigns and educational programs may effectively promote healthy practices among mothers, highlighting the importance of early bonding and neonatal care.

The odds of practicing SSC were 37% lower among women in rural areas compared to those in urban settings. This contrasts with previous studies in Gambia [30], where rural women practiced SSC more. This discrepancy may be due to differences in cultural practices, healthcare access, and support for SSC. Additionally, providers may promote SSC as a cost-free solution to clothing and thermal care shortages in rural areas. Understanding these contextual differences is essential for developing targeted interventions to promote early maternal-infant SSC for all mothers.

Country or region was significantly associated with SSC practices. The odds of practicing SSC were 2.45 times higher in Kenya, 7.23 times higher in Rwanda, and 3.14 times higher in Tanzania compared to Madagascar, highlighting substantial regional variations influenced by healthcare policies, cultural beliefs, and maternal education. In Madagascar, understaffing and limited access to health facilities hinder the use of skilled birth attendants, contributing to low SSC rates [31], Local health workers can play a crucial role through ANC counseling and education, especially for cost-free practices like SSC and breastfeeding. It's important not to focus solely on resource limitations; enhancing non-resource care is vital for improving neonatal health. Understanding these differences is essential for tailoring interventions that promote SSC, ultimately improving maternal and infant health outcomes across the region.

## Conclusion and recommendation

To address the challenges in promoting skin-to-skin contact (SSC), it is essential to increase awareness and training through public health campaigns. Educating healthcare providers and families about the significance of SSC can transform cultural perceptions and encourage its adoption. Strengthening healthcare infrastructure, particularly in rural areas, is vital to ensure that hospitals and clinics are equipped to support SSC effectively. Incorporating SSC as a standard practice in national maternal and neonatal care guidelines will further enhance its acceptance. Local health workers can significantly contribute through peer counseling and support networks for new mothers. Integrating SSC education into antenatal and postnatal programs ensures that both mothers and healthcare providers understand its importance before delivery. By tackling these barriers and promoting SSC, we can improve neonatal health outcomes and reduce mortality rates in East Africa.

### Implication

This study underscores the critical importance of skin-to-skin contact (SSC) in neonatal care while revealing its low prevalence and influencing factors in East Africa. The findings highlight the necessity for targeted interventions to overcome cultural and systemic barriers to SSC. By equipping healthcare providers with this information, the research advocates for public health campaigns and training initiatives that can promote SSC as a standard practice. Implementing these strategies has the potential to enhance neonatal health outcomes and reduce mortality rates among vulnerable populations.

### Acknowledgments

We extend our gratitude to the DHS programmers for granting us access to the pertinent DHS data for this study.

### Author contributions

**Conceptualization:** Alemneh Tadesse Kassie, Tadesse Tarik Tamir, Alebachew Ferede Zegeye.

**Data curation:** Alemneh Tadesse Kassie, Agnche Gebremichael, Daniel Asefa Gonete, Alebachew Ferede Zegeye.

**Formal analysis:** Alemneh Tadesse Kassie.

**Funding acquisition:** Alemneh Tadesse Kassie.

**Investigation:** Alemneh Tadesse Kassie, Astewil moges Bazezew, Daniel Asefa Gonete, Alebachew Ferede Zegeye.

**Methodology:** Alemneh Tadesse Kassie, Tadesse Tarik Tamir, Astewil moges Bazezew, Daniel Asefa Gonete, Alebachew Ferede Zegeye.

**Project administration:** Alemneh Tadesse Kassie.

**Resources:** Alemneh Tadesse Kassie, Astewil moges Bazezew, Alebachew Ferede Zegeye.

**Software:** Alemneh Tadesse Kassie, Tadesse Tarik Tamir, Daniel Asefa Gonete, Alebachew Ferede Zegeye.

**Supervision:** Alemneh Tadesse Kassie, Agnche Gebremichael.

**Validation:** Alemneh Tadesse Kassie, Tadesse Tarik Tamir, Agnche Gebremichael, Daniel Asefa Gonete, Alebachew Ferede Zegeye.

**Visualization:** Alemneh Tadesse Kassie, Astewil moges Bazezew, Agnche Gebremichael, Daniel Asefa Gonete.

**Writing – original draft:** Alemneh Tadesse Kassie, Alebachew Ferede Zegeye.

**Writing – review & editing:** Alemneh Tadesse Kassie, Daniel Asefa Gonete, Alebachew Ferede Zegeye.

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
