## [Decision Letter · Decision Letter 0]

PGPH-D-25-00644

Mother and Newborn Skin to Skin Contact in East Africa: Prevalence and Predictors.

Dear Dr. Kassie,

Thank you for submitting your manuscript to PLOS Global Public Health. After careful consideration, we feel that it has merit but does not fully meet PLOS Global Public Health’s publication criteria as it currently stands. Therefore, we invite you to submit a revised version of the manuscript that addresses the points raised during the review process.

Your submission has been reviewed by two reviewers, whose comments are provided below. Please pay particularly attention to the requests from reviewer 2, and note that any suggestions to cite previously-published work of the reviewers are optional based on the relevance of the suggestions.

We look forward to receiving your revised manuscript.

Kind regards,

Jennifer Tucker, PhD

Staff Editor

Journal requirement:

1. We noticed you have some minor occurrence of overlapping text with the following previous publication(s), which needs to be addressed:

https://pubmed.ncbi.nlm.nih.gov/39664655/

https://www.icsti.org/

In your revision ensure you cite all your sources (including your own works), and quote or rephrase any duplicated text outside the methods section. Further consideration is dependent on these concerns being addressed.

Comments from PLOS Editorial Office: We note that one or more reviewers has recommended that you cite specific previously published works. As always, we recommend that you please review and evaluate the requested works to determine whether they are relevant and should be cited. It is not a requirement to cite these works. We appreciate your attention to this request.

Additional Editor Comments (if provided):

Reviewers' comments:

Reviewer's Responses to Questions

**Comments to the Author**

1. Does this manuscript meet PLOS Global Public Health’s publication criteria? Is the manuscript technically sound, and do the data support the conclusions? The manuscript must describe methodologically and ethically rigorous research with conclusions that are appropriately drawn based on the data presented.

Reviewer #1: Yes

Reviewer #2: Yes

2. Has the statistical analysis been performed appropriately and rigorously?

Reviewer #1: Yes

Reviewer #2: No

3. Have the authors made all data underlying the findings in their manuscript fully available (please refer to the Data Availability Statement at the start of the manuscript PDF file)?

Reviewer #1: Yes

Reviewer #2: Yes

4. Is the manuscript presented in an intelligible fashion and written in standard English?

Reviewer #1: Yes

Reviewer #2: No

5. Review Comments to the Author

Reviewer #1: Abstract

1. The time of the study should be stated in the title

2. The title and purpose should be the same

3. Explain more about Methods (tools),

Introduction

The purpose of the study should be stated at the end of the introduction. B

Methods

1. Explain about Tools "validity and reliability"

Discussion

The discussion is long.

You can use these articles in your introduction and discussion

1. Khadivzadeh T, Karimi FZ, Tara F. Effects of early mother-neonate skin-to-skin contact on the duration of the third stage of labor: A randomized clinical trial. The Iranian Journal of Obstetrics, Gynecology and Infertility, 2018; 21(2): 23-29. doi: 10.22038/ijogi.2018.10704

2. Karimi ZF, Abdollahi M, Khadivzadeh T, Yas A. Investigating the Effect of Kangaroo Mother Care on Maternal-Infant Attachment: A Systematic Review and Meta-Analysis Study, Current Women`s Health Reviews 2024; 20(2) : e280223214099 . https://dx.doi.org/10.2174/1573404820666230228093256

Reviewer #2: This is an interesting manuscript about early skin-to-skin contact in East Africa.

It needs language revisions and some statistic revision (see comment 4b, 7, and 10), and interpretation of results (see comment 8 and 9). There are no line numbers in the manuscript.

Comments:

1. Abstract: Please consider which are the most important results. In the abstract you mention Kenya, Mozambique, and Tanzania. In the main text you mention Rwanda (with the highest prevalence) and Madagascar.

2. Introduction: You have a type error. WHO recommends that skin-to-skin contact should begin within one hour of birth (not at least one hour after).

3. Page 7: All abbreviations in tables (also in the title of the tables) should be explained.

4. Study variables page 8:

a. The age categories are not the same as in Table 2.

b. You have listed “Initiation of first breastfeeding within one hour” as an independent variable, but breastfeeding comes after skin-to-skin contact and is not independent of skin-to-skin contact.

5. Results: There is no need for repeating all results from the tables.

6. Table 3:

a. Please correct the type error in the first Age category.

b. Community level of poverty: Does “low” mean low level of poverty (or low income, which is the opposite)? Please clarify in the table.

7. Page 16: Do mothers in East Africa initiate breastfeeding first (infant wear clothes) and there after initiate skin-to-skin contact? In my country it is opposite (Infants are placed skin-to-skin immediately after birth and will initiate breastfeeding after 30-90 minutes). Provided infants in your study initiated skin-to-skin contact before breastfeeding, you should re-word your results because the odds of SSC is not dependent of breastfeeding, but the odds of breastfeeding is dependent of SSC. In my opinion the following sentence is wrong: “The analysis revealed that the odds of SSC were 2.24 times higher among newborns who did not receive breastfeeding within the first hour compared to those who did (AOR = 2.24, 95% CI: 1.86, 2.69).”

8. Page 16, your sentence: “Additionally, newborns of mothers with only primary education had 1.60 times higher odds of SSC compared to those whose mothers had no formal education or secondary education and above (AOR = 1.60, 95% CI: 1.24, 2.07).” In the variable you have one category as reference, that means you are comparing only to this category (not to “secondary education and above”).

9. Same problem with antennal care visits: You are only comparing to “<4 visits”, not to “no visits”.

10. Regression analyses: Please give a rational in the methods when you choose the middle category as reference instead of highest or lowest (Antenatal care visits, and House hold wealth index).

11. Discussion page 20: Your sentence: “Health service coverage, the quality of maternal and child healthcare, and economic and health policies in Vietnam and the U.S. are generally more robust than those in East African countries.” Please explain why a free of charge intervention like skin-to-skin contact could be influenced by economic.

12. The conclusion is very good. Please also consider to include the role of the health care providers in the discussion, not just in the conclusion.

6. PLOS authors have the option to publish the peer review history of their article (what does this mean?). If published, this will include your full peer review and any attached files.

**Do you want your identity to be public for this peer review?** For information about this choice, including consent withdrawal, please see our Privacy Policy.

Reviewer #1: No

Reviewer #2: No

---

## [Decision Letter · Decision Letter 1]

PGPH-D-25-00644R1

Mother and Newborn Skin to Skin Contact in East Africa: Prevalence and Predictors.

Dear Dr. Kassie,

Thank you for submitting your manuscript to PLOS Global Public Health. After careful consideration, we feel that it has merit but does not fully meet PLOS Global Public Health’s publication criteria as it currently stands. Therefore, we invite you to submit a revised version of the manuscript that addresses the points raised during the review process.

Please address the minor comment raised by reviewer 2. Please also carefully check the manuscript for any lingering typos, such as "The first one hours after birth" -> "The first one hour after birth" (or "The first hour after birth"). 

We look forward to receiving your revised manuscript.

Kind regards,

Sarah Jose, Ph.D.

Staff Editor

Journal Requirements:

1. We noticed you have some minor occurrence of overlapping text with the following previous publication(s), which needs to be addressed:

https://pubmed.ncbi.nlm.nih.gov/39664655/

https://www.icsti.org/

In your revision ensure you cite all your sources (including your own works), and quote or rephrase any duplicated text outside the methods section. Further consideration is dependent on these concerns being addressed.

Additional Editor Comments (if provided):

Reviewers' comments:

Reviewer's Responses to Questions

**Comments to the Author**

1. If the authors have adequately addressed your comments raised in a previous round of review and you feel that this manuscript is now acceptable for publication, you may indicate that here to bypass the “Comments to the Author” section, enter your conflict of interest statement in the “Confidential to Editor” section, and submit your "Accept" recommendation.

Reviewer #1: (No Response)

Reviewer #2: (No Response)

2. Does this manuscript meet PLOS Global Public Health’s publication criteria? Is the manuscript technically sound, and do the data support the conclusions? The manuscript must describe methodologically and ethically rigorous research with conclusions that are appropriately drawn based on the data presented.

Reviewer #1: (No Response)

Reviewer #2: Yes

3. Has the statistical analysis been performed appropriately and rigorously?

Reviewer #1: (No Response)

Reviewer #2: Yes

4. Have the authors made all data underlying the findings in their manuscript fully available (please refer to the Data Availability Statement at the start of the manuscript PDF file)?

Reviewer #1: (No Response)

Reviewer #2: Yes

5. Is the manuscript presented in an intelligible fashion and written in standard English?

Reviewer #1: (No Response)

Reviewer #2: Yes

6. Review Comments to the Author

Reviewer #1: Corrections have been made.

Reviewer #2: Thank you for addressing all my comments.

I have only one comment left: In line 52, the added statement is not consistent with the results in line 270-271 and in Figure 2.

7. PLOS authors have the option to publish the peer review history of their article (what does this mean?). If published, this will include your full peer review and any attached files.

**Do you want your identity to be public for this peer review?** For information about this choice, including consent withdrawal, please see our Privacy Policy.

Reviewer #1: No

Reviewer #2: No

---

## [Editor Report · Decision Letter 2]

Mother and Newborn Skin to Skin Contact in East Africa: Prevalence and Predictors.

PGPH-D-25-00644R2

Dear Lecturer Kassie,

We are pleased to inform you that your manuscript 'Mother and Newborn Skin to Skin Contact in East Africa: Prevalence and Predictors.' has been provisionally accepted for publication in PLOS Global Public Health.

Best regards,

Julia Robinson

Executive Editor